# A Simple, Semi-Quantitative Acyl Biotin Exchange-Based Method to Detect Protein S-Palmitoylation Levels

**DOI:** 10.3390/membranes13030361

**Published:** 2023-03-21

**Authors:** Valentina Buffa, Giorgia Adamo, Sabrina Picciotto, Antonella Bongiovanni, Daniele P. Romancino

**Affiliations:** 1Institute for Research and Biomedical Innovation (IRIB), National Research Council (CNR), Via Ugo La Malfa, 153-90146 Palermo, Italy; 2Integrare UMR_S951 Genethon, Inserm, University of Evry, Université Paris Saclay Genethon, 91000 Evry, France

**Keywords:** membrane protein, post-translational lipidation, S-palmitoylation, acyl biotin exchange (ABE)

## Abstract

Protein S-palmitoylation is a reversible post-translational lipidation in which palmitic acid (16:0) is added to protein cysteine residue by a covalent thioester bond. This modification plays an active role in membrane targeting of soluble proteins, protein–protein interaction, protein trafficking, and subcellular localization. Moreover, palmitoylation is related to different diseases, such as neurodegenerative pathologies, cancer, and developmental defects. The aim of this research is to provide a straightforward and sensitive procedure to detect protein palmitoylation based on Acyl Biotin Exchange (ABE) chemistry. Our protocol setup consists of co-immunoprecipitation of native proteins (i.e., CD63), followed by the direct detection of palmitoylation on proteins immobilized on polyvinylidene difluoride (PVDF) membranes. With respect to the conventional ABE-based protocol, we optimized and validated a rapid semi-quantitative assay that is shown to be significantly more sensitive and highly reproducible.

## 1. Introduction

Protein S-palmitoylation, also called protein thioacylation or S-acylation, consists of the attachment of a 16-carbon fatty acid palmitoyl group to cysteine residue of proteins through a thioester linkage. This post-translational modification is involved in different cellular processes, such as: membrane association of soluble proteins, protein–protein interaction, protein trafficking, subcellular localization, specific membrane domain protein targeting, changing, and modulating protein structure and stability [1,2]. Differently from other lipid modifications, S-palmitoylation is a reversible protein modification [3,4] that is catalyzed by specific palmitoylating enzymes, the DHHC (Asp-His-His-Cys-rich domain) palmitoyl acyl transferase (PAT) family [1,5,6], and depalmitoylating enzymes including palmitoyl esterase, some thiosterases, such as APT1, APT2 (Acyl Protein Thioesterase), and PPT1, PPT2 (Protein–Palmitoyl Thioesterase 1) [5,7]. Protein palmitoylation is related to pathologies such as neurodegenerative diseases, cancer, and developmental defects [7]. Various relevant proteins are palmitoylated into the cell, such as the Ras family of small GTPases, H-Ras and N-Ras oncogenic proteins [8,9], members of the Src family (i.e., LcK, Fyn and Yes) [10] and tetraspanin proteins (e.g., CD151, CD9, and CD63) [11]. Moreover, inside small extracellular vesicles (sEVs), lipid bilayer organelles contain proteins, nucleic acids, and lipids that are exchanged by the cells to establish a cell–cell communication system; protein palmitoylation of the six-transmembrane epithelial antigen1 and 2 (STEAP1, STEAP2), multi-drug resistance-associated protein 4 (ABCC4), transglutaminases, Alix, CD9, and CD63 plays a relevant role for vesicle secretion and cargo distribution [12,13,14].

Recently, interest in developing innovative techniques to detect protein palmitoylation has increased. The most widely used methods for detecting S-palmitoylation are: (i) radioactive methods that rely on metabolically labeling proteins in cells using radioactive-isotope-labeled palmitic acid [5,11,15]; (ii) more sensitive methods based on metabolic labeling with click chemistry [16], and (iii) Acyl Biotin Exchange (ABE) chemistry [5,17,18]. Metabolic labeling with non-radioactive methods takes place through the incorporation of ω-alkynyl-palmitate analog into cellular palmitoylated proteins, its subsequent detection using a copper-catalyzed cycloaddition (click chemistry) of azido-tag (i.e., azido-biotin), and streptavidin–bead isolation of biotinylated proteins [17,19,20]. For the ABE-based method, the detection of all palmitoylated proteins is permitted by a three-step procedure that irreversibly replaces the thioester-linked palmitoyl modifications with stable biotin tags, at neutral pH. However, biotin binding can be reversible if a cleavable HPDP-biotin derivative is used. ABE is achieved by treating cellular lysate or purified proteins (i.e., immunoprecipitated proteins) with methanethiosulfonate (MMTS) or N-ethylmaleimide (NEM); these are alkylating agents that block free sulfhydryl groups. A specific hydroxylamine (HAM) treatment then breaks off the palmitoyl chains from the thioester bonds leaving free cysteine thiol groups, which are biotinylated by a thiol-reactive biotin reagent (i.e., BMMC-biotin, 1-Biotinamido-4-[4′-(maleimidomethyl)cyclohexanecarboxamido]butane) [17]. In vivo metabolic labeling with ^3^H-palmitic acid does not give any information about the palmitate protein’s identity and requires a long labeling time for the cells to obtain a sufficiently detectable signal from the incorporated palmitate [17]. Turnover rates of palmitate labeling also influence detection, because low levels of incorporated palmitate or protein affect the signal level [21,22]. More sensitive detection using ω-alkynyl-palmitate analog and click chemistry bypassed these problems, but potential issues tied to low-abundant expressed proteins or low levels of labeled palmitate incorporation into cells, persist [20]. Moreover, the radiolabeled and ω-alkynyl palmitic acid probes operate via metabolic labeling, which interferes with cell metabolism and may alter normal cell processes. While the ABE-based method can enrich and detect palmitoyl proteins from complex samples, it can also detect false-positive proteins such as native non-palmitoylated thioesters (i.e., the two subunits of mitochondrial pyruvate dehydrogenase, Pdx1 and Lat1) and false assignments of palmitoylation instead of other modifications (i.e., farnesylation or ubiquitination). However, this approach is largely used to study both static and dynamic S-palmitoylation of proteins in native tissues or cell lysates [6,23,24]. So far, different ABE protocols have been proposed in which all steps (NEM and HAM treatments, and biotinylation) can be carried out on lysate in solution, using sequential cycles of chloroform–methanol precipitation, or on a bead–antibody–antigen complex [12,17,22,24,25]. This approach could be problematic in terms of accuracy and in quantitative studies due to the loss of palmitoyl in breaking thioester linkage during long sample preparation. Some of those problems have been bypassed by substituting multiple precipitation steps with the chemical scavenging of NEM by 2,3-dimethyl-1,3-butadiene, which improves sensitivity and accuracy while reducing the time required for the assay [26]. Moreover, some alternative methods such as Acyl-RAC have been developed, based on capturing endogenously S-acylated proteins using thiol-reactive Sepharose beads [27]. Edmonds et al. [28] used both ABE and acyl-resin-assisted capture; this study showed generally good agreement between the two methods, but also pointed out that many identifications were unique to one method, indicating that at least some of the variability is due to methodological differences.

To overcome these limitations, we have developed an innovative and semi-quantitative ABE-based procedure performed on proteins immobilized on protein-binding membranes (e.g., PVDF membranes). We have used CD63 as a model protein to test the efficiency of our protocol. Our approach detects palmitoylation in purified complexes or native proteins from immunoprecipitation or precipitation protocols, avoiding high degradation and/or the loss of proteins in solution.

## 2. Experimental Section

### 2.1. Materials and Reagents

IGEPAL^®^ CA-630, Tween-20, HEPES (4-(2-hydroxyethyl)-1-piperazineethanesulfonic acid), bovine serum albumin (BSA), hydroxylamine solution 50 wt % in H_2_O (HAM), N-ethylmaleimide 98% (NEM), dimethyl sulfoxide (DMSO) Hibry-Max^®^, 2-bromohexadecanoic acid (2BP), and ethylenediaminetetraacetic acid (EDTA), and DL-Dithiothreitol (DTT) were purchased from Sigma-Aldrich, Darmstadt, Germany. Precast 8–16% Precise^TM^ Tris Glycine Gels, Pierce ECL Western blotting substrate, EZ-Link^TM^ BMCC-biotin, Dynabeads protein G, Pierce horseradish peroxidase (HRP)-conjugated streptavidin, and goat anti rabbit Alexa Fluor 680 were purchased from Thermo Scientific, Waltham, MA, USA. Anti-CD63 (H-193) rabbit polyclonal was obtained from Santa Cruz Biotechnology Inc., Dallas, TX, USA. while rabbit polyclonal anti-CD63 was acquired from Invitrogen. PBS (Dulbecco’s Phosphate Buffered Saline) was purchased from Carlo Erba Reagents, Milan, Italy.

### 2.2. Extraction of Proteins from C2C12 Cells

The C2C12 mouse myoblast cell line was grown in high-glucose Dulbecco’s Modified Eagle Medium (DMEM) with 15% fetal calf serum (FCS), containing sodium pyruvate, L-glutamine, penicillin, and streptomycin (Euroclone S.p.A.); it was maintained at 37 °C in a humidified atmosphere with 5% CO_2_. Myotube differentiation was induced with 2% horse serum in place of FCS. After 2 days of differentiation, cells were treated for 1 day with or without 20 μM of 2-bromopalmitate (2BP), a protein palmitoylation inhibitor [12,29,30], in DMEM with 2% horse serum. A concentration of 20 μM 2BP was lower than the 100 μM utilized in other cell lines; this variation was used to counteract the cell apoptosis that occurs in the C2C12 cell line using higher 2BP concentrations [25]. On the third day, cells were washed twice in ice cold phosphate-buffered saline (PBS) and lysed using (0.6 mL/10 cm dish, about 4 × 10^6^ cells) lysis buffers (0.5% Triton X100, 50 mM Tris-HCl, pH 7.5, 5 mM NaCl, 2 mM CaCl_2_, 3 mM KCl, 1 mM MgCl_2_, 100 mM Sucrose (**buffer T**)). The cells were also supplemented with a complete protease inhibitor cocktail without EDTA (Roche, Basel, Switzerland), phosphatase Inhibitor cocktail 2–3 (Sigma), and freshly prepared 10 mM NEM (leaving the ethanol stock solution in agitation at 4 °C for at least 5 min before use). The cell lysate was passed 8–9 times through a 26½ gauge syringe and then gently shaken for 30 min at 4 °C. The cell lysate was centrifuged at 16,000× *g* for 30 min at 4 °C. Clear supernatant was collected and protein concentration was measured using the Pierce BCA protein assay kit (Thermo Scientific). The volume of lysate samples was corrected to the same protein concentration (2 mg/mL).

### 2.3. Immunoprecipitation and Gel Electrophoresis

To prevent nonspecific protein–bead interaction, a pre-clearing step was performed. Thirty microliters of Protein G magnetic beads, pre-equilibrated with lysis buffer, were added to 0.5 mL of lysate samples (with a protein concentration of 0.5–1 mg/mL), and left under agitation for 1 h at room temperature (RT).

Pre-cleared cellular lysates of DMSO (control) and 2BP-treated cells were separated from the magnetic beads, added to 3 μg of CD63 (H-193) antibody, and incubated by rocking overnight at 4 °C. The next day, 30 μL of magnetic beads was added to the samples and left under agitation for 2 h at 4 °C, then washed five times with ice cold buffer containing 0.5 % Triton X100, 50 mM Tris-HCl, pH 7.5, 150 mM NaCl, 2 mM CaCl_2_, 1 mM MgCl_2_ (**IP wash buffer**). 

This procedure was indicated as immunoprecipitation in native/mild conditions. In contrast, when we executed the same washes in 0.5% IGEPAL^®^ CA-630, 50 mM HEPES, pH 7.3, 150 mM NaCl (**buffer A**), we indicated these as stringent conditions. Triton X100 is more hydrophilic; thus, its buffer is less stringent than IGEPAL CA-630.

After the last wash, 45 μL of 2x SDS-PAGE non-reducing (without dithiothretiol, DTT), sample buffer (125 mM Tris-HCl, pH 6.8, 4% SDS, 20% glycerol, and 0.004% bromophenol blue) were added to the beads. The beads were well mixed by vortex, collected by centrifugation, and heated for 5 min at 75–80 °C. In reducing conditions, DTT was used at 5 mM, and may be increased until 10 mM and heated up to 100 °C for 5 min without thioester bonds breakage [15]. Samples were separated from the magnetic beads, transferred into new 1.5 mL tubes, and their volumes determined. DMSO and 2BP-treated samples were split into two parts, loaded onto a 10% sodium dodecyl sulfate-polyacrylamide gel electrophoresis (SDS-PAGE) and electrophoresed.

After electrophoresis, separated proteins were transferred overnight at 4 °C, with a constant current of 0.09 A, onto 0.2 μm PVDF (Amersham™ Hybond™ P, GE Healthcare Life Science, Little Chalfont, PA, USA) or nitrocellulose membrane (Amersham Protran Premium 0.2NC, GE Healthcare Life Science). The membranes were stained with Ponceau S (Sigma-Aldrich) for 2 min, visible lanes were suitably cut, and strips were separately placed on a 4-well non-treated rectangular Nunc™ dish (Thermo Fisher Scientific). The strips were then decolored with 2 washes of 10 min each in PBS at RT, and lastly in PBS. Optionally, the process can be stopped at this stage and membranes stored in PBS at 4 °C in the fridge for up to 2 days (the membranes must always be kept moist).

### 2.4. ABE onto PVDF Membranes and in Solution

All steps of the ABE procedure were performed, with a gentle rocking at room temperature (RT). The membranes were washed 3 times with 3 mL of either **buffer A** or **buffer B** (0.1% IGEPAL^®^ CA-630, 50 mM HEPES, pH 7.3, 150 mM NaCl) for 10 min (**step 1**); incubated in 3 mL of either buffer A or B containing 0.1% SDS (stringent buffer) for 10 min, (**step 2**); washed 3 times with buffer A for 10 min (**step 3**) and; incubated in 3 mL of buffer A supplemented with freshly prepared 65 mM NEM for 2 h (**step 4**). During this step, a HAM buffer (1 M HAM, 50 mM HEPES, pH 7.3, 150 mM NaCl, 0.5% IGEPAL^®^ CA-630) was created. To obtain a 1 M concentration of HAM, a proper volume of hydroxylamine solution was added to a new tube containing buffer A, and a pH meter was used to adjust the pH to exactly 7.3 (the range 7.2–7.4 was permitted). Subsequently, the membranes were each incubated twice in 3 mL of buffer A for 10 min to remove NEM residues (**step 5**). At this point, we removed all buffer A from wells and treated the membranes separately with either 3 mL of HAM buffer (+HAM samples) or the same volume of buffer A (−HAM samples) for exactly 1 h (**step 6**). While step 6 was carried out, a stock solution of biotin was made. We weighed 1 mg of BMCC-biotin and added it to 2.4 mL of DMSO in a new tube, then mixed and completely solubilized by rocking in the dark at RT to obtain a clear 0.8 mM BMCC-biotin solution. Biotin buffer was completed by mixing 3.75 μL of 0.8 mM BMCC-biotin stock solution with 6 mL of buffer C (50 mM HEPES, pH 6.2, 150 mM NaCl, 0.5% IGEPAL^®^ CA-630). After step 6, all the membranes were washed twice with buffer C, and supplemented once with 1 mM EDTA for 10 min and twice with buffer C for 10 min (**steps 7–8**); residue of buffer C was then removed from wells and incubated with 3 mL of biotin buffer for each membrane for 1 h (**step 9**). Lastly, membranes were washed 5 times with TBS for 20 min (**step 10**). At this stage, the membranes could be treated or stored in PBS or TBS at 4 °C in the fridge for up to 2 days.

ABE in solution was performed in Eppendorf tubes as described by Brigidi et al. [22]. After incubation of the immunoprecipitated samples using CD63 (H-193) antibody for 2 h at 4 °C, magnetic beads were sequentially washed in agitation. They were washed once with 0.6 mL of buffer A (pH 7.5), 10 mM NEM; samples were then divided into two parts: 0.2 mL (–HAM control) and 0.4 mL (+HAM treatment). The two parts were integrated with buffer A (pH 7.5), 10 mM NEM, at the final volume of 0.5 mL, incubated at 0 °C for 10 min, quickly washed once in ice cold 0.5 mL of stringent buffer (pH 7.5), and washed 3 times in 0.5 mL of buffer A (pH 7.2), without NEM. They were then incubated in 0.5 mL of HAM buffer and buffer A (pH 7.2), as control, for 1 h at RT; washed once in 0.5 mL of buffer C; incubated in 0.5 mL of biotin buffer for 1 h; washed once in 0.5 mL of buffer C; and 3 times in buffer A. At this point, the palmitoylated proteins were biotinylated, and samples were prepared and loaded on PAGE-SDS.

A 3% BSA blocking buffer (not powdered milk) in TBS with 0.05% Tween-20 (TBST) for 2 h was used to block the PVDF membranes; after 2 washes with TBST for 5 min, the membranes were incubated with HRP-conjugated streptavidin diluted 1:20,000 in blocking buffer for 1 h. Subsequently, the membranes were washed once with blocking buffer for 30 min, and 4–5 times with TBST for 30–60 min. To detect the palmitoylated protein bands, the membranes were incubated for 1 min using a chemiluminescent substrate (Pierce ECL Western blotting substrate, Thermo Scientific), and then exposed for 10 sec to 5 min to Amersham Hyperfilm™ ECL (GE Healthcare Life Science).

### 2.5. Western Blotting

After identification of S-palmitoylation, the membranes coming from both ABE in PVDF and in the solution protocol were tested to detect CD63 by adapting Western blotting to Odyssey detection. The blots were probed with anti-CD63 (H-193) diluted 1:200 with blocking solution [Odyssey^®^ blocking buffer (LI-COR) diluted 1:1 with TBST, 50 mM TrisHCl, 150 mM NaCl, 0.05% Tween^®^20 (TBST)] for 2 h at RT on a rocking shaker, washed 4–5 times with TBST for 30 min, and incubated with Alexa Fluor 680 goat anti-rabbit IgG (Invitrogen) diluted 1:5000 with blocking solution for 1 h at RT. Afterwards, the membranes were washed 4–5 times with TBST for 30 min and 3 times in TBS for 5 min; all washes were performed at RT. To determine signal intensity from infrared CD63 bands, Odyssey Infrared Imaging System (LI-COR) and LI-COR imaging software were used. Alternatively, after stripping the membranes using a stripping buffer (i.e., 0.1 M Glycine, pH 1.8 for 5–10 min at RT), conventional Western blotting was completed using rabbit polyclonal anti-CD63 (H-193) as primary antibodies, anti-rabbit HRP conjugated as secondary antibodies, and a chemiluminescent substrate (Pierce ECL Western blotting substrate, Thermo Scientific) in conjunction with autoradiography on Amersham Hyperfilm™ ECL (GE Healthcare Life Science, Little Chalfont, PA, USA) for signal detection.

### 2.6. Statistical Analysis

Data were expressed as mean ± S.D. and were evaluated using Student’s *t* test. Mean differences were considered statistically significant when *p* values were less than 0.05 (*) or 0.005 (**).

## 3. Results and Discussion

### 3.1. Assay Design

The purpose of this study was to set up a straightforward and reliable ABE procedure to detect a specific palmitoylated protein when compared to their unmodified counterpart. Our procedure could be easily adapted to analyze the palmitoylation level in different palmitoyl proteins, either enriched by an immunoprecipitation (IP) or native co-immunoprecipitation (Co-IP) step. We focused on the tetraspanin protein CD63, known to be palmitoylated in different cell types [11,31]. The general methodology is diagrammed in Figure 1. The immunoprecipitated proteins were separated on PAGE-SDS and electrophoretically transferred onto PVDF membranes. Subsequently, all free sulfhydryl groups were blocked by incubation with NEM, the fatty acyl group was either removed using HAM (+HAM lanes), or not as control (−HAM lanes); a final incubation with a thiol-specific reagent (i.e., BMMC-biotin) allowed the labeling of the free sulfhydryl groups. Afterwards, we detected the biotin-tagged palmitoylation sites with streptavidin–HRP and ECL Western blot and, lastly, the target protein with immunoblot detection (Odyssey detection).

### 3.2. Setup of Detergent Concentrations in ABE Buffers and Membranes

One potential drawback of an ABE-based assay is the incomplete blockage of all free sulfhydryl groups on the proteins of interest. We surmise that the extent of sulfhydryl alkylation is influenced by exposed sulfhydryl groups (e.g., type and detergent concentration in the buffer composition). A previously described ABE methodology [22] successfully utilized IGEPAL CA-630 to extract palmitoylated proteins and included it in the ABE buffers. Thus, we first optimized the IGEPAL CA-630 concentration in the ABE buffers used in our assay. Total protein lysates of C2C12 cells were separated by PAGE-SDS, electrophoretically transferred onto protein-binding membranes (i.e., PVDF) and ABE-tested in the presence of different concentrations of IGEPAL^®^ CA-630 detergent [22]. As shown in Figure 2a, in contrast to buffers without detergent, the best results in terms of clear detection of palmitoylated bands were obtained when we utilized a concentration of 0.1–0.5% IGEPAL^®^ CA-630 in the buffers. We also tested 1% IGEPAL^®^ CA-630 and 0.05% Tween-20 (data not shown) in the buffers, with the results mostly overlapping with 0.5% IGEPAL^®^ CA-630 in terms of sharper bands. Moreover, when we used nitrocellulose membranes as an alternative to PVDF, we observed (Figure 2b) less accurate results in terms of the intensity, clear detection, and number of palmitoylated bands. Thus, the hydrophobicity of PVDF membranes makes them an ideal support for our ABE applications. Proteins are tightly bound, and are quantitatively retained during exposure to acidic, basic, or organic solvents (i.e., ABE buffers).

### 3.3. Efficient Blockage of Unspecific Free Sulfhydryl Groups Using NEM as an Alkylating Agent

A potential problem of the ABE procedure is preventing the presence of unspecific free sulfhydryl groups that could interact with BMCC-biotin and cause false positive bands. Firstly, we prevented breakage of palmitic acid-cysteine bonds on protein samples using a low concentration of the mild reducing agent DTT (5 mM final concentration) and boiling them for 5 min, before loading them onto PAGE-SDS. This treatment left the thioester bonds of palmitoylated proteins unchanged for the subsequent detection [15]. Afterwards, to block free sulfhydryl groups of proteins on the PVDF membranes from IP with CD63 (H-193), we tested two different concentrations, 20 and 65 mM, of the alkylating agent NEM (Figure 3). We considered it suitable to use NEM at 65 mM final concentration for at least 120 min at room temperature in PVDF membranes, as it resulted in lower levels of nonspecific bands in the −HAM lane compared to 20 mM of NEM at the end of treatments (see −HAM lane in Figure 3). This result was in accordance with previous studies reported by Drisdel et al. [5], which experimentally determined the incubation times and NEM concentration needed to block free sulfhydryls in a cell lysate.

Immunoprecipitation in native conditions from total extracted proteins (1 mg) with CD63 antibody (3 μg) and washing steps occurred as described in the experimental section. After PAGE-SDS and transfer onto PVDF membranes, the membranes were incubated for 2 h using different NEM concentrations. Palmitoylated CD63 bands were identified using HRP-conjugated streptavidin and ECL detection. Exposure time to Amersham Hyperfilm was the same for both experiments.

### 3.4. Comparison of ABE Protocols 

In order to compare our ABE protocol on membranes to the previously described ABE in solution [22], we performed a preliminary immunoprecipitation in native/mild conditions using a C2C12 cell lysate and CD63(H-193) antibodies, as described in the experimental section. We then performed the ABE on PVDF membranes in parallel with the protocol in solution described by Brigidi et al. [22]. The ABE on PVDF was carried out according to the protocol described in the experimental section.

To carry out the ABE in solution, after the same immunoprecipitation step described above, antigen–antibody complexes from control and 2BP treated cells were washed and handled as described [22] in the experimental section.

In Figure 4, the complex pattern of palmitoylated proteins obtained from ABE on PVDF in stringent conditions revealed 60, 50, and 42 kDa CD63 bands [32] (Figure 4(a1)), while in native/mild conditions they showed an additional 37 kDa CD63 band (Figure 4(a2)). Analysis of the palmitoylation/CD63 protein ratio in Figure 4(a3) reveals a decrease of about 30% of quantifiable 42 kDa bands in the 2BP-treated samples, when the ABE on PVDF was performed in stringent conditions (see Figure 4(a1)); it also reveals a decrease of about 30% in the 2BP-treated samples of two other quantifiable bands, 50 and 37 kDa, obtained on PVDF in native/mild conditions (see Figure 4(a2)). The 60 kDa band was not quantifiable for the palmitoylation/CD63 protein ratio because this band was embedded with other CD63 bands ranging from about 80 to 60 kDa; likewise, in native/mild conditions, the other most prominent palmitoylated band, shown at 42 kDa (Figure 4(a2)), could not be precisely quantified for its CD63 palmitoylation level due to the strong band reaction in these conditions. The palmitoylated bands ranging from 160 to about 70 kDa present in Figure 4(a1,a2) probably belong to the CD63 protein or to CD63-interacting proteins that are palmitoylated. Additionally, semi-quantitative results were obtained by immunoprecipitation and ABE experiments carried out with another protein, Alix, in our laboratory, using this procedure on PVDF [12].

In contrast, when ABE in solution was carried out, only a protein band of 42 kDa (Figure 4(b1)) was detected in these conditions, and it was not possible to establish an obvious decrease in the palmitoylation level of these bands after 2BP treatments (Figure 4(b2)).

These results suggest that the ABE procedure on PVDF is quantitative, at least for the 50, 42 and 37 kDa CD63 bands, and showed sensitivity more than or similar to ABE in solution [22].

Moreover, this protocol could be applied to detect and quantify palmitoylated proteins that may interact with the studied protein in Co-IP experiments.

## 4. Conclusions

The ABE procedure on PVDF is a successful, simple, and sensitive assay for detecting palmitoylated proteins. Importantly, this semi-quantitative test permits the analysis of the amount of palmitoylated proteins after different treatments, such as using palmitoylation inhibitors. Moreover, the versatility of IP, coupled with our ABE test, could allow researchers to analyze the presence/level of palmitoylation in any proteins interacting with the investigated protein.

## Figures and Tables

**Figure 1 membranes-13-00361-f001:**
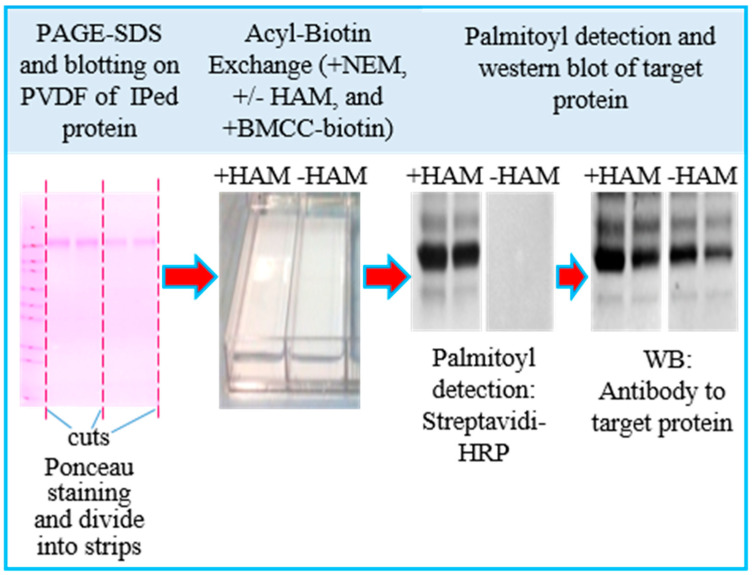
Schematic diagram of the ABE procedure.

**Figure 2 membranes-13-00361-f002:**
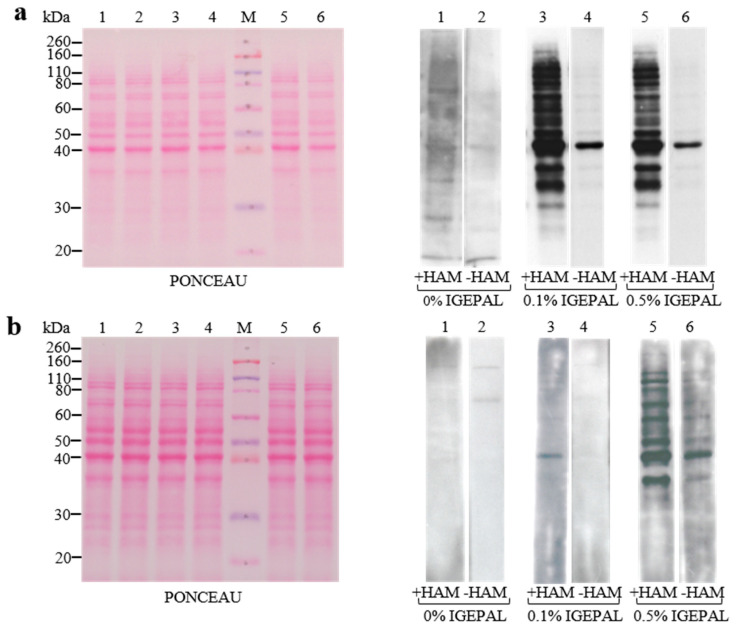
Comparison of ABE protocols using different membranes and IGEPAL CA-630 concentrations. Whole lysates from C2C12 cells (25 µg) were electrophoresed on PAGE-SDS, transferred onto PVDF (**a**) and onto nitrocellulose (**b**), Ponceau S stained, and analyzed by the ABE procedure using different IGEPAL^®^ CA-630 concentrations. After the ABE protocol on the membranes, palmitoylated bands were identified using HRP-conjugated streptavidin and ECL detection. The same exposure time to Amersham Hyperfilm in ECL detection was set for the blots in (**a**,**b**).

**Figure 3 membranes-13-00361-f003:**
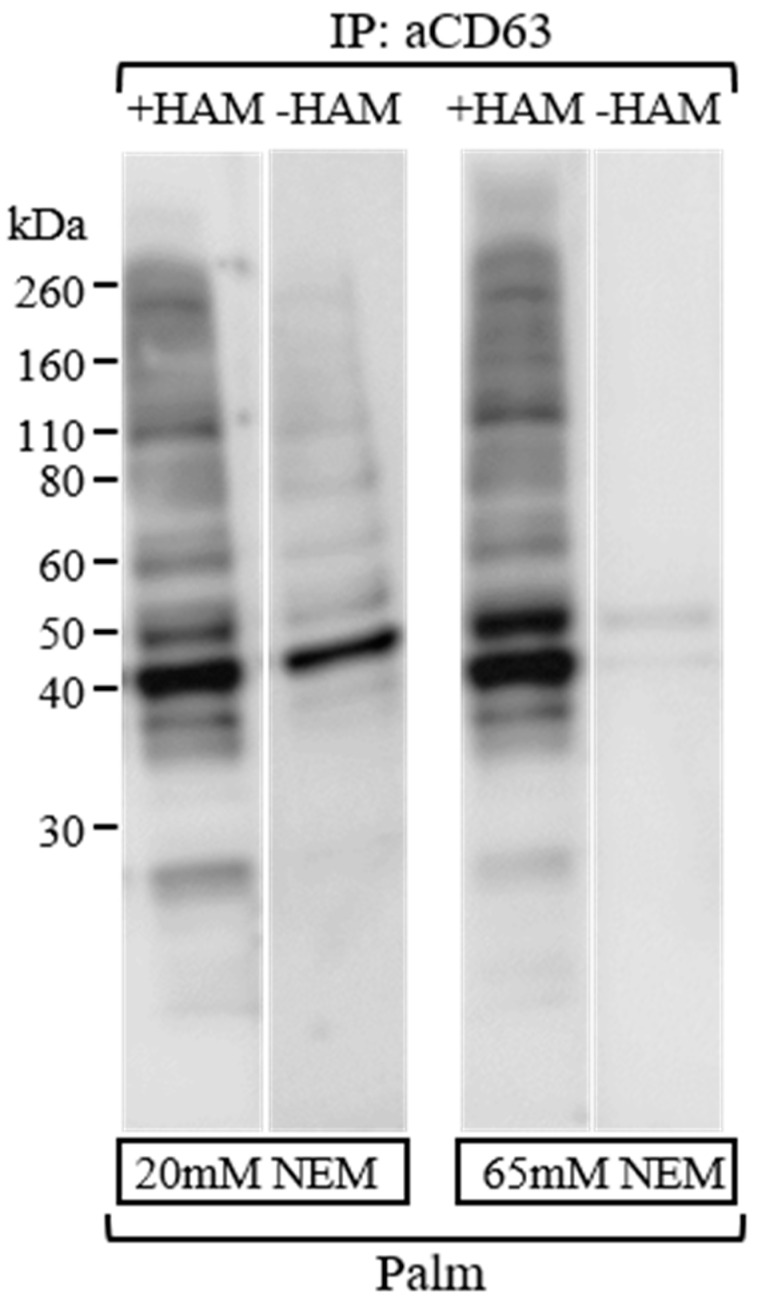
Comparison of ABE protocols using different NEM concentrations.

**Figure 4 membranes-13-00361-f004:**
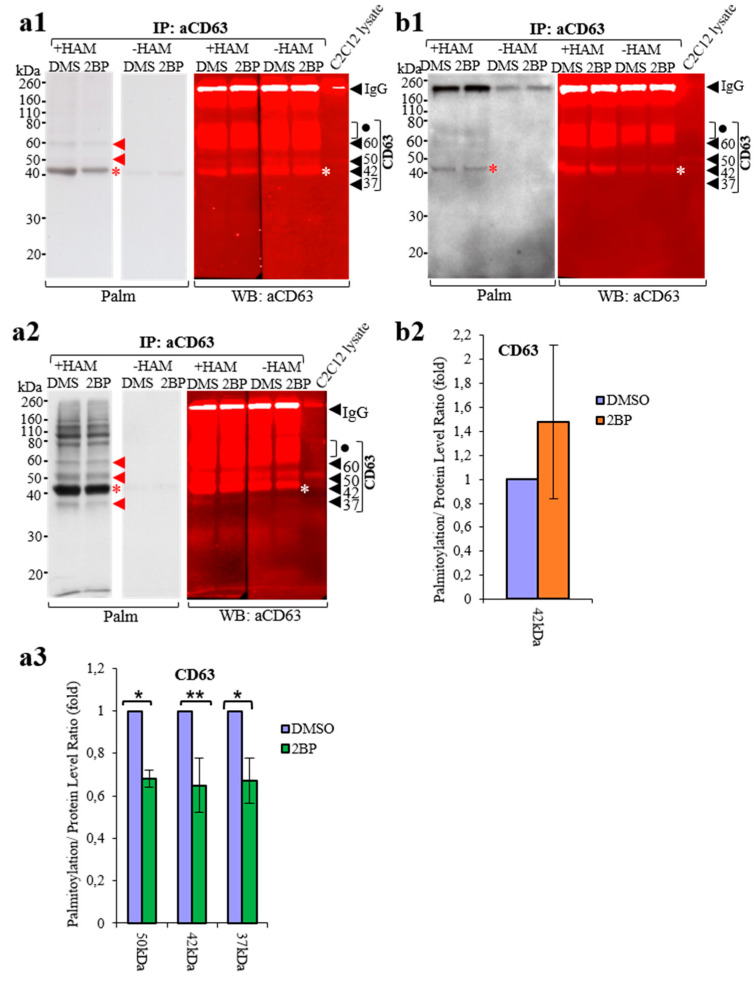
Comparison of ABE using different protocols. DMSO and 2BP indicate the lanes containing IP from the control and 2BP-treated C2C12 cells. PVDF membranes containing the same pair of DMSO and 2BP immunoprecipitated samples were resuspended with nonreducing sample buffer and loaded. The ABE protocol on PVDF (**a**), which came from PAGE-SDS loaded with IP using the CD63 (H193) rabbit polyclonal antibody described in the experimental section, is compared to the ABE protocol carried out on the immunoprecipitated complexes, using the procedure in solution [22] (**b**). IP washes and other common procedures in both experiments, (**a**,**b**), were performed equally. The experiments in (**a2**) were performed as in (**a1**) using a less stringent IP wash in native/mild conditions (see experimental section). Red and white * indicate a prominent palmitoylated 42 kDa band of CD63 that was strongly visible in all experiments (**a1**,**b1**,**a2**). Red and black arrows indicate, respectively, palmitoylated and CD63 bands, while dots indicate CD63 bands faintly seen in the C2C12 lysate. Bands of 50, 42 and 37 kDa CD63 were quantified in the histogram (**a3**) as the palmitoylation/protein level ratio for DMSO and 2BP treatments, and were referred to experiments performed in (**a2**), relative to 50 and 37 kDa bands, or in (**a1**), for the 42 kDa band. The histogram (**b1**) shows a palmitoylation/protein-level ratio obtained from experiments performed as in (**b1**) using the Brigidi et al. [22] protocol. All data are shown as mean +/− SD of five independent experiments. Data are expressed relative to the control (DMSO) set at 1. * indicates *p* < 0.05, while ** indicates *p* < 0.005 vs. untreated control (DMSO).

## Data Availability

The Data presented in this study are contained within the article.

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
