# Peer review of "A Simple, Semi-Quantitative Acyl Biotin Exchange-Based Method to Detect Protein S-Palmitoylation Levels"

_membranes, 2023, doi:10.3390/membranes13030361_

Round 1

Reviewer 1 Report

The authors developed a new method for the detection of protein S-palmitoylation on PVDF membranes, which may be interesting to readers in the field. However, several issues need to be concerned.

1. The problem is a new assay is that the PVDF membrane has to be separated into two parts for sequential steps with and without HAM, it is difficult to compare the signals from two PVDF membranes. The shortcomings should be mentioned and discussed in the manuscript.

2. DTT was used in the sample preparation, even at the concentration which the authors used, the S-palmitoylation may be destroyed. Thus, the controls with different concentrations of DTT need to be added.

3. Another question is why there are so many bands in the IP sample of aCD63, the specificity of the antibody may be verified.

4. The author used 2-BP to detect the effect of S-palmitoylation, but the effect seems not significant, it is much better to use an S-palmitoylation site mutation protein as a control. In addition, at least one more substrate should be tested in the system to establish this new method.

5. English writing needs to be improved.

Author Response

Response to Reviewer 1 Comments

Point 1: The problem is a new assay is that the PVDF membrane has to be separated into two parts for sequential steps with and without HAM, it is difficult to compare the signals from two PVDF membranes. The shortcomings should be mentioned and discussed in the manuscript.

Response 1. The PVDF membrane was cut after the blotting of the same gel, and the two parts were treated the same way. The only difference was that the “control membrane” was treated with the same buffer but without HAM. Moreover, all data were analysed as a palmitoylation/protein level ratio to eliminate problems of possible lack of accuracy. We mention this in the manuscript.

Point 2: DTT was used in the sample preparation, even at the concentration which the authors used, the S-palmitoylation may be destroyed. Thus, the controls with different concentrations of DTT need to be added.

Response 2. DTT was not used in the experiment shown in Figure 4 that ultimately showed the detection level of the ABE-membrane protocol vs. the ABE-in solution one. We used DTT in the experiment in Figure 2 to evaluate the different membranes and IGEPAL CA-630 concentrations. However, as described by Schmidt et al. 1988, a low level of DTT was almost compatible with this assay. Nevertheless, we used the same low DTT level as published by Brigidi et al. 2013.

Point 3: Another question is why there are so many bands in the IP sample of aCD63, the specificity of the antibody may be verified.

Response 3.The many bands present in the IP sample of aCD63 are due to different highly N-glycosylated forms of CD63, as described by Tominaga et al. 2014.

Point 4: The author used 2-BP to detect the effect of S-palmitoylation, but the effect seems not significant, it is much better to use an S-palmitoylation site mutation protein as a control. In addition, at least one more substrate should be tested in the system to establish this new method.

Response 4. We use 2-BP at a concentration of 20 mM in these experiments and in others included in the paper by Romancino et al. 2018, because we evaluated that this is the maximum concentration that does not induce cell apoptosis in the C2C12 cell line. In C2C12, the effect of 20 mM was significant (about 40%), though not total, and sufficient to test the detection level of our method.

Point 5: English writing needs to be improved

Response 5. As suggested by Reviewer 1 we have now improved the English by extensive revision of the text by a native English-speaking editor.

Reviewer 2 Report

The Authors describe a modification of the detection technique of S-palmitoylation of proteins, which consists in substituting thioester-bonded acyl residues with biotin, so-called ABE. In its classical version, the ABE technique is technically demanding because the exchange of individual reaction mixtures requires multiple protein precipitations. The method proposed in the manuscript consists in carrying out the ABE reaction on the PVDF membrane after prior immunoprecipitation of a protein and its separation in SDS-PAGE. The washing of membrane-bound samples significantly facilitates conducting the ABE procedure.

The work is very interesting, but the Authors should prove more convincingly that the method they propose has a chance of success.

1. The blots shown in Fig. 4, especially the detection of fluorescently labeled CD63 and visualized with the Odyssey, are of low quality. The reason is probably too much material applied to the gel or antibodies were not diluted enough. Samples with a better resolution of the CD63 protein should be shown. The detection of palmitoylation in Fig. 4a1 seems most strong, a detection with lower concentration of biotin should be done to quantify the results for 42 kDa band.

2. The result of the ABE procedure in the lysate is not convincing. For 7 replicates, the deviation in the BPA-treated sample is too big. Again, the detection of the CD63 protein by the fluorescently labeled antibody can be questioned. Using during immunoblotting anti-CD63 of another host than those used as the primary and secondary antibody for immunoprecipitation will reduce or even abolish the signal generated by IgG. Although the protocol of Brigidi and Bamji (2013) recommends doubling the volume of +HAM sample vs -HAM, I do not think this is correct. You can see that the -HAM sample is less abundant in CD63 and IgG which indicates that the amount of the protein in +HAM and -HAM samples was not equalized. Does the covalent attachment of BMCC-biotin affect the migration of CD63 in the gel? I propose to abandon this part of the work or improve it.

3. In its place, I recommend showing results of the new ABE procedure applied for another protein. ALIX is a good choice. Equally good choices may be, e.g.,  flotillin 1 and 2, against which very good antibodies are available. These are cytosolic or submembranous proteins, which excludes problems with protein solubilization, as may be the case with the transmembrane protein CD63.

To strengthen their case, the Authors can show dose-dependence, i.e., apply the protein in a given range of amounts. S-palmitoylation of flotillins is very sensitive to BPA inhibition. The Authors can also take into account that BPA prepared in complexes with BSA is not toxic to cells, can be used at higher concentrations, and more efficiently inhibit protein S-palmitoylation.

The manuscript is well written, except for the description of Figure 4 (see below). There are some points to be addressed:

-          Some information on CD63 immunoprecipitation is missing: what was the host of anti-CD63 antibody, and what antibody was linked to magnetic beads?  Why 0.5% Triton is mild and 0.5% IGEPAL harsh?

-          In many places a strange symbol is seen instead of Greek letters. Check the text carefully, at least 10 such symbols can be found.

-        Page 3: “While, for ABE-based method the detection of all palmitoylated proteins was permitted by a three step procedure that irreversibly replaces the thioester-linked palmitoyl modifications with stable biotin tags”.

Biotin binding can be reversible if a cleavable HPDP-biotin derivative is used. In fact, the original version of ABE technique requires the application of such biotin to subsequently release the protein from biotin-streptavidin beads in reducing conditions. Correct the sentence accordingly.

-    Page 6: At present the description on the technique ends on the preparation of HAM-treated membranes.

Was the detection of biotinylated protein conducted as described for ABE in solution? This has to be clearly stated, since the incubation with streptavidin and chemiluminescence it the real finish point of the proposed procedure.

-          Page 7: After, at RT. the membranes were washed 4-5 times with TBST for 30 min, 3 times in TBS for 5 min,.. Correct this sentence.

-     Fig. 3: After, PAGE-SDS and transfer on PVDF membrane, the ABE  protocol was executed using different NEM concentration, as indicated in figure, for 2 hours. Correct this sentence.

-          Fig. 3. “Expo indicates the exposure time to Amersham Hyperfilm in ECL detection”. There is no such information in the Figure.

-      Page 10: Analysis of palmitoylation/CD63 protein ratio in Figure 4a2 reveals a quantifiable 42kDa band when the ABE on pvdf was done in stringent condition (Figure 4a) that decrease about 30%, and others 2 quantifiable bands, of 50 and 37 kDa on pvdf in native condition (Figure 4a1), that also decrease about 30% (Figure 4a).

This sentence is hard to understand. What are native conditions? Does native mean mild? When the reduction is observed? Does it concern samples treated with BPA? The whole description of Fig 4a1,2,and3, most important to the manuscript is weak an chaotic.

Also Fig. 4 legend should be improved, there are some repetitions, like the citation of Ref 22.

-        We can find biotin-BMCC and BMCC-biotin, biotin with low and capital letters. Unify to BMCC-biotin.

-        We can find PVDF and pvdf, unify to PVDF.

-        Odissey- Odyssey is the correct name of the apparatus.

 Comments also in the attached file review (1).docx

Author Response

Responses to Reviewer 2 Comments

Point 1: The blots shown in Fig. 4, especially the detection of fluorescently labeled CD63 and visualized with the Odyssey, are of low quality. The reason is probably too much material applied to the gel or antibodies were not diluted enough. Samples with a better resolution of the CD63 protein should be shown. The detection of palmitoylation in Fig. 4a1 seems most strong, a detection with lower concentration of biotin should be done to quantify the results for 42 kDa band.

Response 1. We thank the reviewer for this suggestion. The blots shown in Figure 4 are representative of different experiments and of different exposures of the same immunoblots; we believe that the immunoblot shown in Fig.4a1 permits the visualization and quantification of the 37 kDa band. The detection of palmitoylation for the 42 kDa band was done in Fig.4a, because, as suggested by Reviewer 2, in Fig. 4a1 it is too strong to be quantified. Furthermore, the histograms in Fig. 4a2 and b1 are the mean plus SD of five independent experiments.

Point 2: The result of the ABE procedure in the lysate is not convincing. For 7 replicates, the deviation in the BPA-treated sample is too big. Again, the detection of the CD63 protein by the fluorescently labeled antibody can be questioned. Using during immunoblotting anti-CD63 of another host than those used as the primary and secondary antibody for immunoprecipitation will reduce or even abolish the signal generated by IgG. Although the protocol of Brigidi and Bamji (2013) recommends doubling the volume of +HAM sample vs -HAM, I do not think this is correct. You can see that the -HAM sample is less abundant in CD63 and IgG which indicates that the amount of the protein in +HAM and -HAM samples was not equalized. Does the covalent attachment of BMCC-biotin affect the migration of CD63 in the gel? I propose to abandon this part of the work or improve it.

Response 2. We thank the reviewer for highlighting the high deviation in the 2BP-treated samples, as determined by the ABE-in solution protocol (Fig. 4b and b1). Indeed, this feature of the ABE-in solution protocol rendered it, in our view, of low reproducibility. For this reason we looked for and set up the ABE-in membrane protocol that we propose as an alternative to the ABE-in solution protocol as it has higher reliability from one experiment to another (Fig. 4a, a1 and a2). As suggested by Review 2, for the immunoblotting we have used an anti-CD63 of another type rather than those used as the primary and secondary antibodies for immunoprecipitation, thus reducing the signal generated by IgG and detecting only a 150 kDa band of IgG, in no reducing condition, far from CD63 bands that does not interfere with the CD63 quantification. In our experiment, we used a double volume of +HAM sample vs -HAM because we referred to Brigidi et al. 2013, and although in Fig. 4a1 we can see that the -HAM sample is less abundant in CD63 and IgG (which indicates that the amount of the protein in +HAM and -HAM samples was not equalized in Fig. 4a1) in Fig. 4a, the situation is the contrary. However, all data were analysed as a palmitoylation/protein level ratio to eliminate problems of possible lack of accuracy. I think that a difference of only 0.53 kDa (BMCC-biotin weight) for residue, after covalent attachment of BMCC-biotin, does not change the migration of CD63 in SDS-PAGE very much.

Point 3: In its place, I recommend showing results of the new ABE procedure applied for another protein. ALIX is a good choice. Equally good choices may be, e.g.,  flotillin 1 and 2, against which very good antibodies are available. These are cytosolic or submembranous proteins, which excludes problems with protein solubilization, as may be the case with the transmembrane protein CD63.To strengthen their case, the authors can show dose-dependence, i.e., apply the protein in a given range of amounts. S-palmitoylation of flotillins is very sensitive to BPA inhibition. The Authors can also take into account that BPA prepared in complexes with BSA is not toxic to cells, can be used at higher concentrations, and more efficiently inhibit protein S-palmitoylation.

Response 3. We thank the reviewer for this insightful suggestion. In line with his/her comment, we have shown the results of the ABE-in membrane protocol for the Alix protein in a previous paper (Romancino et al. 2018). In addition, we used 2-BP at a concentration of 20 mM in these experiments and in others included in the paper by Romancino et al. 2018, because we evaluated that this is the maximum concentration that does not induce cell apoptosis in the C2C12 cell line. In C2C12, the effect of 20 mM was significant (about 40%), though not total, and sufficient to test the detection level of our method.

Other Points: Some information on CD63 immunoprecipitation is missing: what was the host of anti-CD63 antibody, and what antibody was linked to magnetic beads?  Why 0.5% Triton is mild and 0.5% IGEPAL harsh?

Response for other points. As suggested by Reviewer 2, we have included the host of anti-CD63 in the text. The antibody linked to magnetic beads was CD63 (H-193) rabbit polyclonal as indicated in Results and discussion, “Comparison of ABE protocols”, line 337 and inEfficient blockage of unspecific free sulfhydryl groups using NEM as alkylating agent”, line 316. 0.5% Triton X100 is milder than 0.5% IGEPAL, because it is slightly more hydrophilic and less stringent (https://www.snowpure.com/docs/triton-x-100-sigma.pdf).

In many places a strange symbol is seen instead of Greek letters. Check the text carefully, at least 10 such symbols can be found.

We thank the Reviewer for bringing this issue to our attention; accordingly, we have corrected this symbol in the text.

Page 3: “While, for ABE-based method the detection of all palmitoylated proteins was permitted by a three step procedure that irreversibly replaces the thioester-linked palmitoyl modifications with stable biotin tags”. Biotin binding can be reversible if a cleavable HPDP-biotin derivative is used. In fact, the original version of ABE technique requires the application of such biotin to subsequently release the protein from biotin-streptavidin beads in reducing conditions. Correct the sentence accordingly.

We have added a sentence in the text as suggested by Reviewer 2 in the page 3.

Page 6: At present the description on the technique ends on the preparation of HAM-treated membranes. Was the detection of biotinylated protein conducted as described for ABE in solution? This has to be clearly stated, since the incubation with streptavidin and chemiluminescence it the real finish point of the proposed procedure.

We thank the Reviewer for bringing this omission to our attention. As suggested, we have now added a sentence in the text to clarify this issue.

Page 7: After, at RT. the membranes were washed 4-5 times with TBST for 30 min, 3 times in TBS for 5 min,.. Correct this sentence.

We have corrected the sentence.

Fig. 3: After, PAGE-SDS and transfer on PVDF membrane, the ABE  protocol was executed using different NEM concentration, as indicated in figure, for 2 hours. Correct this sentence.

We have corrected this sentence.

Fig. 3: “Expo indicates the exposure time to Amersham Hyperfilm in ECL detection”. There is no such information in the Figure.

The caption of Fig. 3 is now corrected.

Page 10: Analysis of palmitoylation/CD63 protein ratio in Figure 4a2 reveals a quantifiable 42kDa band when the ABE on pvdf was done in stringent condition (Figure 4a) that decrease about 30%, and others 2 quantifiable bands, of 50 and 37 kDa on pvdf in native condition (Figure 4a1), that also decrease about 30% (Figure 4a).This sentence is hard to understand. What are native conditions? Does native mean mild? When the reduction is observed? Does it concern samples treated with BPA? The whole description of Fig 4a1,2,and3, most important to the manuscript is weak an chaotic. Also Fig. 4 legend should be improved, there are some repetitions, like the citation of Ref 22.

We thank the Reviewer for bringing this issue to our attention; the descriptions and legends of Fig. 4 and 3 are now improved.

We can find biotin-BMCC and BMCC-biotin, biotin with low and capital letters. Unify to BMCC-biotin.

We have corrected the text and unified it to BMCC-biotin.

We can find PVDF and pvdf, unify to PVDF.

We have corrected the text and unified it to PVDF.

Odissey- Odyssey is the correct name of the apparatus.

Yes, the complete name is Odyssey Infrared Imaging System (LI-COR Biosciences) as indicated in the experimental section.

Reviewer 3 Report

S-palmitoylation of cellular proteins is involved in a broad array of physiological processes. Given the importance of proteins that undergo palmitoylation (and function as regulators of transport, signal transduction, etc.) understanding their function, and in particular their regulation, is of high priority. In their manuscript entitled “A simple, semi-quantitative acyl biotin exchange-based method to detect protein S-palmitoylation levels” Buffa and collaborators attempt to establish an alternative protocol for one of the most widely used methods for palmitoylation detection. The methodological approach is in general well designed and properly conducted while the obtained results confirm that there is indeed a potential in what the authors offer. On the other hand, a broader discussion would be required in light of some alternative methods such as Acyl-PEG exchange or Acyl-RAC (Tewari et al. 2020 J Vis Exp. 10: 10.3791/61016). For example, Edmonds et al. (2017 Sci Rep 7: 3299) used both acyl-biotin exchange and acyl-resin-assisted capture approaches showing generally good agreement between the two methods, but also pointing that many identifications were unique to one method. There are also a few reports on substantial improvements of the acyl-biotin exchange methods, including Hurst et al. 2017 BioTechniques 62: 69, where the authors reported that substituting multiple precipitation steps with chemical scavenging of N-ethylmaleimide by 2,3-dimethyl-1,3-butadiene enables to greatly improve sensitivity and accuracy while reducing time required for the assay. Moreover, Woodley et al. (2019 Methods Mol Biol 1977:71) recently adapted this method to permit global S-acylation site analysis. This protocol, when combined with SILAC-based quantification, allowed both the large-scale identification of palmitoylation sites and profiling of palmitoylation site changes. While embedding their report within the current trends, the authors would also have more possibilities to expose the selling point of their work. There also a few further issues listed below which should be addressed by the authors.

- The quality of some images of PVDF, in particular these presented on Figure 4, is not sufficient. The CD63 bands in WB seem to be very heterogeneous. What are the bands between 160 and 260 kDa in 4b, panel “Palm”? …they are not visible in other panels. Does the palmitoylation/protein level ratio shown on 4b1 corresponds to what can be seen on 4b (maybe more representative image could be chosen to confirm it)? It also seems that 42 kDa band is not the only one that corresponds to CD63 and could be seen on 4b and (compare with l. 354-355).

- All four panels within Figure 1 should have the same layout. It is also not fully clear what the sentence “Expo indicate the exposure time to Amersham Hyperfilm in ECL detection.” (within Figure 2 legend) refers to.

- It should be clearly stated whether electrotransfer was performed in the same way for both (in solution vs on membrane) approaches (see l. 206)      

- there are numerous typos (e.g. symbol for “micro” is lost throughout the whole text, missing or redundant spaces and commas, l. 375, ref. 15 with lost numbering, etc.) or grammar (e.g. l. 224) errors which should be corrected. Moreover, there are some inconsistencies in abbreviations (e.g. SDS-PAGE vs PAGE-SDS, PVDF vs pvdf)

Author Response

Response to Reviewer 3 Comments

Point 1: S-palmitoylation of cellular proteins is involved in a broad array of physiological processes. Given the importance of proteins that undergo palmitoylation (and function as regulators of transport, signal transduction, etc.) understanding their function, and in particular their regulation, is of high priority. In their manuscript entitled “A simple, semi-quantitative acyl biotin exchange-based method to detect protein S-palmitoylation levels” Buffa and collaborators attempt to establish an alternative protocol for one of the most widely used methods for palmitoylation detection. The methodological approach is in general well designed and properly conducted while the obtained results confirm that there is indeed a potential in what the authors offer. On the other hand, a broader discussion would be required in light of some alternative methods such as Acyl-PEG exchange or Acyl-RAC (Tewari et al. 2020 J Vis Exp. 10: 10.3791/61016). For example, Edmonds et al. (2017 Sci Rep 7: 3299) used both acyl-biotin exchange and acyl-resin-assisted capture approaches showing generally good agreement between the two methods, but also pointing that many identifications were unique to one method. There are also a few reports on substantial improvements of the acyl-biotin exchange methods, including Hurst et al. 2017 BioTechniques 62: 69, where the authors reported that substituting multiple precipitation steps with chemical scavenging of N-ethylmaleimide by 2,3-dimethyl-1,3-butadiene enables to greatly improve sensitivity and accuracy while reducing time required for the assay. Moreover, Woodley et al. (2019 Methods Mol Biol 1977:71) recently adapted this method to permit global S-acylation site analysis. This protocol, when combined with SILAC-based quantification, allowed both the large-scale identification of palmitoylation sites and profiling of palmitoylation site changes. While embedding their report within the current trends, the authors would also have more possibilities to expose the selling point of their work. There also a few further issues listed below which should be addressed by the authors.

Response 1.We thank the reviewer for the positive comments about this work and for his/her helpful suggestions. Accordingly, we have included some relevant references and discussion in the introduction.

Point 2: The quality of some images of PVDF, in particular these presented on Figure 4, is not sufficient. The CD63 bands in WB seem to be very heterogeneous. What are the bands between 160 and 260 kDa in 4b, panel “Palm”? …they are not visible in other panels. Does the palmitoylation/protein level ratio shown on 4b1 corresponds to what can be seen on 4b (maybe more representative image could be chosen to confirm it)? It also seems that 42 kDa band is not the only one that corresponds to CD63 and could be seen on 4b and (compare with l. 354-355).

Response 2.The bands present between 160 and 260 in Fig. 4b in Palm panel was an unspecific reaction due to the high amount of IgG heavy chain; the same reaction isn’t visible in other Palm panels probably because the amount of IgG was lower. The palmitoylation/protein level ratio of CD63 42 kDa was very heterogeneous in the ABE-in solution protocol (Fig. 4b and b1), and was not statistically significant. The 42 kDa is not the only one that corresponds to CD63 and can be seen in 4b, such as the immunoblot in 4b shown, but Palm signals were not detected for CD63 between 80 and 60 kDa in those panels, probably due to the low level of protein or, alternatively, due to the presence of hidden palmitoylation sites on HAM during the ABE-in solution protocol.    

Point 3: All four panels within Figure 1 should have the same layout. It is also not fully clear what the sentence “Expo indicate the exposure time to Amersham Hyperfilm in ECL detection.” (within Figure 2 legend) refers to.

Response 3.The Figure 1 and 2 legend have been corrected.

Point 4: there are numerous typos (e.g. symbol for “micro” is lost throughout the whole text, missing or redundant spaces and commas, l. 375, ref. 15 with lost numbering, etc.) or grammar (e.g. l. 224) errors which should be corrected. Moreover, there are some inconsistencies in abbreviations (e.g. SDS-PAGE vs PAGE-SDS, PVDF vs pvdf)

Response 4. We have corrected the text.

Round 2

Reviewer 1 Report

The authors have answered most of the questions.

Author Response

We thank the reviewer for the positive evaluation

Reviewer 2 Report

The Authors improved the manuscript. They should, however, agree on a few questions that they left as comments in the revised version of the manuscript - pages 3 (twice), 5, 9, 14.

Additionally,

Page 5 : The Authors probably meant the Complete Protease Inhibitor Cocktail without EDTA, not… "completely free of EDTA".

Page 13, lane 419 of the legend to Figure 4: "PVDF membranes containing the same pair of DMSO..." the Authors probably meant immunoprecipitates.

Author Response

We thank the reviewer for bringing these errors to our attention. The manuscript is now uploaded in the final form, without any comments and including the corrected paragraphs (pg. 5 and pg.13, legend to Figure 4).

Reviewer 3 Report

I would like to thank the authors for addressing the issues raised by reviewers. The manuscript has been improved substantially, but still seems to be premature for publication. First of all, it is not clear why the authors submitted a file containing comments which do not seem to be addressed to editors and/or reviewers. Secondly, the authors could not offer anything better in regard to the low-quality blots/IPs which in their current form might be misleading for a reader and suggest that the presented approach does not guarantee obtaining reliable results.

Author Response

1) We thank the reviewer for bringing this error to our attention. The manuscript is now uploaded in the final form, without any comments.

2) The blots/IPs shown in the figures are representative of different experiments and of different exposures of the same immunoblots. For instance, the immunoblot shown in Fig.4a1 permits the visualization and quantification of the 37 kDa band of CD63. The detection of palmitoylation for the 42 kDa band was done in Fig.4a, because in Fig. 4a1 it is too strong to be quantified. As a consequence, different level of exposure is shown of representative blot, and the histograms in Fig. 4a2 and b1 are the mean plus SD of five independent experiments. 

Round 3

Reviewer 3 Report

I would like to thank the authors for submitting the corrected version of the manuscript.